# Merkel Cell Polyomavirus (MCPyV) and Cancers: Emergency Bell or False Alarm?

**DOI:** 10.3390/cancers14225548

**Published:** 2022-11-11

**Authors:** Maria Georgia Dimitraki, George Sourvinos

**Affiliations:** Laboratory of Clinical Virology, Medical School, University of Crete, 71003 Heraklion, Crete, Greece

**Keywords:** Merkel cell polyomavirus, MCPyV, Merkel cell carcinoma, MCC, non-MCC, viral carcinogenesis, infectious pathogens, human oncogenic virus

## Abstract

**Simple Summary:**

Merkel cell polyomavirus is a widespread pathogen, strongly associated with the highly aggressive Merkel cell carcinoma. In this review paper we aim to examine a possible association between Merkel cell polyomavirus and the emergence of different types of cancer. Evidently, Merkel cell polyomavirus is the major oncogenic factor for Merkel cell carcinomas, but no establishing results can be derived about the virus’ prevalence and role in the pathogenesis of other malignant diseases. Most certainly, it is imperative that the virus’ biology, infection cycle, as well as its interactions with the host, are further investigated. Identifying a link between a virus and tumorigenesis is crucial and could lead to new virus-targeted therapeutic approaches with one final target: cancer elimination.

**Abstract:**

Merkel cell polyomavirus (MCPyV), the sole member of Polyomavirus associated with oncogenesis in humans, is the major causative factor of Merkel cell carcinoma (MCC), a rare, neuroendocrine neoplasia of the skin. Many aspects of MCPyV biology and oncogenic mechanisms remain poorly understood. However, it has been established that oncogenic transformation is the outcome of the integration of the viral genome into the host DNA. The high prevalence of MCPyV in the population, along with the detection of the virus in various human tissue samples and the strong association of MCPyV with the emergence of MCC, have prompted researchers to further investigate the role of MCPyV in malignancies other than MCC. MCPyV DNA has been detected in several different non-MCC tumour tissues but with significantly lower prevalence, viral load and protein expression. Moreover, the two hallmarks of MCPyV MCC have rarely been investigated and the studies have produced generally inconsistent results. Therefore, the outcomes of the studies are inadequate and unable to clearly demonstrate a direct correlation between cellular transformation and MCPyV. This review aims to present a comprehensive recapitulation of the available literature regarding the association of MCPyV with oncogenesis (MCC and non-MCC tumours).

## 1. Introduction

Cancer is a group of diseases with a major global health impact, estimated to become the leading cause of premature death in this century, according to GLOBOCAN 2020 [1]. The emergence of cancer is primarily linked to both genetic predisposition and environmental factors [2]. Nevertheless, recent studies have identified some infectious agents as risk factors of cancer development in humans [3]. According to data from 2018, infection-attributable cancers represented the third leading cause of cancer development (2.2 million cases) [4], following smoking and diet [5]. Additionally, infection-related cancer cases have mostly been reported in the developing parts of the world [4]. Most of those oncogenic agents are characterized as Group I human carcinogens by the International Agency for Research on Cancer (IARC) and they include viruses, bacteria, and parasites [6,7,8]. Recent advances in the fields of molecular biology have shed light on the correlation between fluke infections and cancer formation and, therefore, have provided valuable insights into the molecular basis of oncogenesis [9]. Generally, infectious agents seem to be able to cause malignant transformation via direct and indirect mechanisms: persistent infection that eventually leads to inflammation, DNA alternations and cell damage, irregular oncogene expression, and immunologic recognition disruption [7,8,9]. Amongst the infectious agents, studies targeting the precise oncogenic mechanisms of the viruses are the most abundant.

Human oncogenic viruses have been the in the centre of scientific attention for over 50 years. They represent a variety of viruses with a range in morphology, genetic material and, overall, viral life cycle and reproduction [4,10]. According to The World Health Organization, approximately 10% of infection-attributed cancers have viral aetiology [11]. The carcinogenic mechanisms of oncoviruses also vary widely [12] but they generally involve continued expression of specific viral oncogenes that regulate proliferative and antiapoptotic activities, dysregulation of cellular genomic instability through integration of the viral DNA into the host genome, viral promotion of DNA damage and immune evasion strategies [10,12]. However, it is significant to note that oncogenesis is an uncommon consequence of the viral infection, and it is developed mainly after a long-lasting chronic infection [10]. Moreover, the detection of a virus in a cancerous tissue does not necessarily establish causality [10]. Therefore, the direct or indirect involvement [13] of suspected viruses in the tumorigenesis process has been proposed to be made through the following list of characteristics: (1) presence and persistence of viral genome in biopsies obtained from tumour tissues; (2) presence of growth-promoting viral genes in model systems; (3) emergence of the malignant tissue exclusively due to continuous viral oncogene expression or dysregulation of host genes; and (4) epidemiological evidence of correlation between viral infection and cancer development [14]. So far, seven viruses have been confirmed to have an established causative link to oncogenesis. These are human papillomaviruses (HPV), hepatitis B virus (HBV), hepatitis C virus (HCV), Epstein–Barr virus (EBV), Kaposi’s sarcoma associated herpes virus (KSHV) (also called human herpes virus 8), human T-cell leukaemia virus (HTLV-1), and Merkel cell polyomavirus (MCPyV) [5,11]. This review focuses on Merkel cell polyomavirus and examines its association with various types of cancer.

## 2. Merkel Cell Polyomavirus and Merkel Cell Carcinoma

### 2.1. Merkel Cell Polyomavirus (MCPyV)

Merkel cell polyomavirus (MCPyV) is the only member of the Polyomaviridae family with scientifically proven ability to cause oncogenesis in humans [15]. MCPyV exhibits the characteristic genomic and structural organization of the family (Figure 1). It is a small, non-enveloped virus, approximately 45 nm in diameter, with icosahedral capsids [16]. The genome is represented by a short (approximately 5.4 kb) double-stranded circular DNA [17], packed together with host-derived histones [18]. It is comprised of three main regions, classified based on the sequential priority in which they are expressed during infection: the early-coding region, the late-coding region, as well as a noncoding, regulatory region (NCRR) [19]. NCRR is located in-between the other two and does not encode any functional protein or RNA. It possesses an essential role in the viral life cycle regulation, since it contains both the viral origin of replication (Ori) and, also, various transcription promoters [20,21]. Immediately upon viral infection, the expression of the early region is initiated, encoding multiple, alternatively spliced RNA transcripts, leading to distinctive gene products that are generally involved in the replication of the viral genome [17]. Those are the large tumor antigen (LT), the small tumor antigen (sT), as well as a 57 kT antigens, along with the alternate LT ORF (ALTO) product [20,21]. The LT, sT and 57 kT antigens all share an identical sequence of 78 amino acids at their N-terminal [22]. Lastly, the viral genome also contains a gene, located on the late strand of the T antigen genes, with an opposite transcription orientation to the T antigens. The transcriptional product of this gene is a non-coding miRNA [20,21]. The most important viral antigens proven to play a vital part in the lytic infection are the MCPyV LT and sT antigens [23]. More specifically, the LT antigen is a multifunctional, pleiotropic protein of 700 amino acids that impacts several cellular proteins involved in both the host cell and the viral life cycles [18,22]. It is mainly nuclear located, though phosphorylation modifications may lead to different localization within the cell [18]. Structurally, it is composed of a highly conserved N-terminal region (CR1) and a C-terminal region with an origin-binding domain (OBD) and a helicase domain, both critical for viral DNA replication. In-between the two terminal regions there can be found sequences responsible for the protein–protein interactions of LT with host proteins that end up alternating the normal cell cycle (RB,p53) [22]. Regarding the sT protein, it shares the LT N-terminal region, but has a unique C-region made of a protein phosphatase 2A (PP2A) binding site [22,24]. Following the expression of the early-coding locus and the replication of the viral DNA, the late region is transcriptionally activated to produce the structural components of the viral capsid, major capsid protein viral protein 1 (VP1) and the minor capsid protein 2 and 3 (VP2 and VP3) [20]. The presence of VP1 and VP2 is crucial for the viral entry into the cell, whereas VP3 cannot be detected in MCPyV-infected cells or in MCPyV virions, indicating a possible conditional expression pattern [18].

In general, MCPyV is a widespread and mostly asymptomatic resident of the skin in the human population [25,26,27]. Serological surveys for antibodies against viral antigens demonstrate that 20–40% of children 0–5 years old test positive [28], meaning that the first contact with this infectious agent occurs early in life [29], whereas positivity reaches 80% in individuals of age of 50 years and above [28,30,31]. Therefore, it is highly suggested that MCPyV is a member of the skin microbiome (virome) of the human body [32]. Moreover, the precise host cell tropism of MCPyV remains unclear. Interestingly, MCPyV seems to require both salicylic acid heparan sulphate upon its entrance [29,32,33], molecules ubiquitous in several cell types, though the replication of MCPyV has so far shown to be limited to keratinocytes [29] and fibroblasts from the dermis, and the lung [34]. Furthermore, among fibroblasts isolated from common model animals, only chimpanzee and human ones can support a vital level of MCPyV replication [35]. This lack of an eligible cell culture system for MCPyV infection [21], as well as the inability to establish an in vivo infection system [35], has limited our ability to understand the fundamentals of the viral behavior. Consequently, there is much to be elucidated with respect to MCPyV host potential reservoir cells and the complete MCPyV infectious cycle during infection. Despite the limitation regarding the target-cell preferences, MCPyV is not solely detectable within the skin, but also in respiratory samples, urine, and blood [20,36], as well as in samples extracted from numerous non-malignant tissues (e.g., spleen, bone marrow, stomach, heart), although with a comparatively low viral load [37]. Lastly, the route of MCPyV transmission has yet to be established, but suggested routes include primarily direct contact with the skin or saliva, and recently proposed ones, i.e., airborne and fecal–oral [18,29].

Nevertheless, MCPyV is not harmless in the minority of individuals and, on the contrary, it is considered the first, and, to date, the only polyomavirus directly implicated in the emergence of an aggressive, lethal human cancer, Merkel cell carcinoma (MCC).

### 2.2. Merkel Cell Carcinoma (MCC)

Merkel cell carcinoma (MCC) is a rare, aggressive, neuroendocrine neoplasia of the human skin [38], originally described in 1972 by Cyril Taker [39]. With a nearly 50% mortality rate [40] and a high case-fatality [21], it is considered one of the most lethal skin malignancies, surpassing melanoma [41]. The low survival rates are a direct consequence of the rapid metastatic ability (distant, regional metastases or lymph node metastases) of MCC [29,40,42,43], combined with its intrinsic ability to resist immunological eradication [29,43], as well as a relatively poorly response to chemotherapeutic agents and constant recurrence [44,45,46]. Furthermore, MCC is characterized by several established risk factors, including advanced age (50 years and older), population features (fair skin), exposure to intense solar radiation (skin displayed mainly to UV rays) and immune deficiencies [42,43,47]. In the last few years, even though MCC is still considered uncommon, both the incidence as well as the mortality rate of MCC have sharply tripled [21] and are anticipated to rise further [48]. Once emerged, MCC appears as a fast growing red or purple nodule of trabecular, nodular, or diffuse shape [47]. Despite the given name, attributed to the phenotypical similarities of the tumour with the Merkel cells of the skin, the true cell origin of MCC is rigorously debated [43,49]. The initial favoured theory, that MCC arises from unrestrained growth of differentiated Merkel cells, is debated due to the post-mitotic nature of the Merkel cells, which makes them terminally differentiated, unable to undergo further cell division, and consequently, making their oncogenic potential limited [20,50]. In addition, Merkel cell have epidermal origin, in contrast to the dermis or subcutis layer origin of the MCCs [20]. Therefore, plenty of other suggestions have been made, such as that MCCs could potentially occur from Merkel cell precursor cells present at the hair follicles of the skin, or other epithelial, fibroblastic, or even B-cells, primarily in accordance with morphological similarities shared by MCC and these potential pedigrees [20,51,52].

The two major causative factors of MCC are Merkel cell polyomavirus (MCPyV) and ultraviolet radiation [41,53]. Hence, Merkel cell carcinomas can be subdivided into two major groups: the virus-associated MCCs (VP-MCCs) and the virus-negative MCCs (VN-MCCs or UV-MCCs). Despite evidently different aetiologies, both groups share mutations in vital cell-cycle related genes (meaning they regulate similar molecular paths in different means) [29,54,55,56], as well as similar disease phenotype [57] and same emergence location (on sun-exposed regions of the body, such as the head, neck, and limbs [58,59]). On the contrary, VP-MCCs seem to have an insignificant mutational burden and a lower number of somatic mutations compared to VN-MCCs [60,61,62,63]. Initially, the MCCs caused by UV-light overexposure have been attributed to approximately 20% of the total cases and are characterized by high accumulation of UV-derived DNA mutations, specifically on genes that encode tumour-specific UV-neoantigens, such as the retinoblastoma (pRb) pathway, RB1, TP53, and PIK3CA, along with mutations in host DDR and chromatin modulation pathways [43,53,60,61]. Such aberrations are crucial for MCC emergence, since the dysregulation results in both interruption of the cell cycle and induction of SOX2 expression, leading to neuroendocrine transformation [64]. MCPyV-negative MCCs also have increased levels of activation-induced cytidine deaminase (AID) which could potentially enhance carcinogenesis [17].

Whilst UV radiation is considered a strong risk factor and leads to MCC-related mutagenesis in VN tumours, the major causative factor of MCC is MCPyV, evident in the vast majority of MCC cases [40]. More specifically, MCPyV was first detected and isolated in 2008 by Feng and colleagues at the Pittsburgh Cancer Institute within the cells of the primary MCC tumour and a metastatic lymph node, using digital transcriptome subtraction assays. The viral genome, which was detected in 80% of the MCC samples, also exhibited a clonal integration pattern within several different chromosomal sites [15,29,40]. This identification of MCPyV was a significant leap in the comprehension of the pathogenesis of MCC in virus-associated MCC, but the exact role of MCPyV in MCC pathogenesis requires further investigation [21]. In the last few years, significant progress in decoding the mechanism of MCC-VP tumorigenesis has been observed. Thus, oncogenic transformation by MCPyV is hypothesized to be the outcome of two consecutive events [29]: (1) the incorporation of the viral DNA into the host genome, (2) the subsequent expression of the viral oncogenic proteins. Initially, the clonal incorporation of MCPyV DNA into the host genome transpires at generally random genomic sites, most commonly on chromosome 5. This event has been evident in up to 80% of all studied MCCs [65,66]. The integration takes place as either a single copy or as a concatemer of multiple copies, always in a manner that results in loss of replicative abilities of the virus before MCC development, but, simultaneously, in a way that preserves the expression of the so-called viral tumour (T) antigens [20,67]. The MCPyV-encoded antigens, sT and LT, are highly immunogenic and are required for the MCC development as well as the oncogenic phenotype maintenance [23,50]. They contribute to oncogenesis by targeting various host cell proteins involved in cell cycle control and proliferation [20]. The significance of the T antigens in MCC tumorigenesis is highlighted by the fact that their expression is required for optimal MCC cell growth and proliferation [9,68]. Depletion of these tumour antigens from VP-MCC cell lines, on the other hand, leads to cancer cell death [20]. Starting with the LT antigen, sequencing assays have indicated several truncating mutations within the C-terminus of LT, most probably caused by and during the viral genome integration process. These mutated LT antigens seem to lack the DNA binding domain–helicase activity, thus interrupting the viral DNA replication in MCC tumours [17]. However, the truncated LT proteins retain several other structural domains and motifs within the N-terminus region, resulting in interaction with the tumour-repressor retinoblastoma protein (pRb) [68], thus inactivating it and promoting cell cycle deregulation and cellular proliferation (Figure 2). MCPyV LT is also associated with Vam6/Vps39-like protein, which is potentially significant for the viral cell cycle [54]. The oncogenic significance of the LT antigen is further indicated by the observation that the silence of this antigen in VP-positive MCC-derived cells intervenes with the cell growth and promotes cell death [17,69]. All in all, the accumulation of mutations in the LT antigen plays a vital role in the carcinogenesis process, since these mutations not only downregulate the viral replication and viral load, promoting immune evasion, but also promote unrestrained cell proliferation. Regarding the sT antigen, it is more frequently detected in human MCC tumours than the LT antigen and is considered a more crucial ‘player’ for oncogenesis [18]. Experiments on transgenic mouse models have indicated a transformative ability of sTs in various organ systems, including in the epidermis [54]. Furthermore, the sT antigen is able to transform rat-1 fibroblasts in cell cultures [54]. ST has been found to interact with a variety of host cellular proteins and, therefore, exhibits a range of activities. Initially, sT’s PP2A region interacts and inhibits PP2A and PP4 phosphate complexes. This interaction promotes inhibition of the nuclear factor-κB (NF-κB), as well as alternations in the cellular cytoskeleton, resulting in cell motility. Nevertheless, the PP2A binding domain is not considered crucial for carcinogenesis, since studies have shown that its presence is not required for cellular transformation by MCPyV ST31 [54]. An sT antigen region that is vital for the virus-induced oncogenesis is the LT-stabilizing domain (LSD), which has been suggested to interact with F-box/WD repeat-containing protein 7 (FBXW7) and cell division cycle protein 20 homologue (CDC20), resulting in inhibition of E3 ubiquitin ligases, and subsequent enhancement of the oncoprotein stability and cap-dependent mRNA translation, respectively. Additionally, research on sT has demonstrated its ability to promote changes in gene expression, by regulating both EP400 histone acetyltransferase and chromatin remodelling complex66 [54]. Lastly, studies suggest that sT might also interfere with the cell metabolism [54]. Therefore, sT antigen appears to possess a substantial role in MCPyV-related cancer emergence. Finally, immunocompromised patients display a higher (16-fold) relative risk for MCC development, opposed to the healthy population [18]. Therefore, immuno-suppression is classified as another crucial contributing factor in MCPyV-mediated carcinogenesis [43,70,71]. It is also imperative to mention that in MCPyV-positive MCCs, UV light may simply promote tumour growth through immunosuppressive effects on the tumour microenvironment [9].

According to molecular epidemiological studies, the prevalence of MCPyV in MCC varies widely depending on the region of interest [5,72], ranging between approximately 25% in Australia and 100% in a French study [72]. Subsequent studies demonstrated values of 76.0% in the United States population and 66.6% in Switzerland [5]. Furthermore, MCPyV DNA has been shown to be present in 55–79% of MCCs in Japanese patients [72]. The detection methods used for MCC diagnosis initiate with clinical examination and are followed by tissue biopsy through immunochemistry, a method that demonstrates characteristic histopathologic neuroendocrine features [43]. Immunohistochemistry aims to differentiate MCCs from other, morphologically similar, neuroendocrine tumours through characteristic staining of a variety of MCC epithelial and neuroendocrine markers, such as cytokeratin-20 (CK20), neurofilaments, CAM 5.2, TTF-1 and AE1/3 [73,74]. With the purpose of distinguishing the VP-MCCs from VN-MCCs, a series of identification techniques are performed, including serological anti-oncoprotein antibody detection and molecular techniques of viral DNA amplification and detection (PCR, qPCR) [75]. For the molecular assays, the most common primers used target the LT and sT gene and the VP1 and NCCR regions of the genome [75]. Lastly, for confirmation purposes, a combination of molecular and immunohistochemical analyses can be performed (since viral proteins can be detected in tumour tissues) [75]. On average, the viral genome copy number of MCPyV has been estimated to be 60 times lower in healthy tissues across the body compared to MCC samples [76]. The available treatment options for both UV- and virus-induced MCCs, are, so far, very limited. Surgical extraction followed by rounds of chemotherapy is the recommended route [43]. Due to the immunogenic properties of MCCs, a novel therapy based on immune checkpoint inhibitors has recently shown encouraging survival outcomes [62,77,78], though this approach is considered insufficient for the systemically immunosuppressed patients [79,80]. Lastly, therapeutic vaccines are currently being developed [9,81].

### 2.3. MCPyV and Non–Merkel Cell Carcinomas

As aforementioned, MCPyV is chronically shed from the skin [32] of the vast majority of the healthy population globally and it has been detected, not solely on the skin, but also in a variety of other tissues, thus raising the possibility that the virus establishes an infectious reservoir in body tissues other than the skin, such as the lymph nodes or within the blood. Accordingly, the widespread prevalence of the virus across the human body in combination with the strong association of MCPyV in the development of MCC have led to the following question in the scientific community: what if MCPyV is present and has an oncogenic linkage to other, non-MCC cancers? This question has prompted researchers to further investigate a possible existence of MCPyV in cancers emerging throughout the body, searching for a possible linkage between malignancies and the MCC-inducing virus. Therefore, cancerous tumours, derived from different body locations, have been systematically tested for the presence of viral DNA, antigen transcripts, and proteins. In most cases, the MCPyV DNA was extracted from the tumour biopsies using PCR-based methods, whereas the expression of the LT antigen was monitored by immunohistochemistry. Interestingly, LT could only be detected in a handful of cases, even though the viral DNA was present [82]. Lastly, it is important to mention that the research projects only investigated the presence of the viral genome and the LT antigens and scarcely the integration of the genome or the truncated LT antigen [82], even though these characteristics are the hallmarks for the MCPyV-positive MCCs.

The outcomes of these investigations are listed below, divided into groups according to the system of the body where each malignancy was isolated (Table 1).

#### 2.3.1. Circulatory System

The presence of MCPyV in tumours has been tested in different types of leukaemia, such as acute myeloid leukaemia (AML), cutaneous T-cell leukaemia (CTCL), cutaneous B-cell lymphomas (CBCL), and chronic lymphocytic leukaemia/small lymphocytic lymphoma (CLL/SLL). Initially, acute myeloid leukaemia (AML) is a type of blood cancer of the myeloid line of blood cells (granulocytes or monocytes) in the bone marrow, which results in interference with normal blood cell production. AML progresses rapidly and is typically fatal in the span of a few weeks if left untreated. Samples of AML cells have been tested using qPCR as well as NGS. From the 29 samples, only one was found positive for MCPyV DNA [83,84]. Furthermore, the results from studies on cutaneous T-cell lymphoma (CTCL), a special non-Hodgkin’s lymphoma caused by a mutation of T cells which migrate to and reside in the skin, do not demonstrate a clear outcome regarding MCPyV. In particular, from the 352 studied samples of CTCL, some contained no detectable levels of MCPyV cells [85,86], while others reported some level of MCPyV DNA [87,88] (viral load 0.0012–12.467 viral copies per cell) [82]. The methods used were PCR/Southern blot and qPCR. Regarding cutaneous B-cell lymphomas (CBCL), slowly growing B-cell lymphomas of the skin, studies detected the virus in CBCLs using PCR/qPCR, with a relatively low prevalence (5.6%) [82], a low viral load per cell, and without viral protein expression (IHC for LT antigen was negative for all samples) [88,89,90]. Another type of B-cell leukaemia/lymphoma is chronic lymphocytic leukaemia/small cell lymphoma (CLL/SLL). It is the most common B-cell-related leukaemia affecting >15,000 patients/year [91,92], and it has an established epidemiological link with Merkel cell carcinoma (a large population-based study revealed that CLL patients have a higher risk of developing VP-MCCs) [92,93]. Therefore, there has been some interest in investigating a possible leukemogenic role of Merkel cell polyomavirus (MCPyV). PCR studies have demonstrated the presence of MCPyV in peripheral blood at low copies [94,95,96,97]. The viral load was only determined in one case, and it was extremely low, with 0.000017–0.002 viral copies per cell [82,96]. Moreover, amongst the positive samples, six contained a truncated LT antigen of which two also harboured full-length LT mRNA [94]. Lastly, no MCPyV genome was detected in chronic myelomonocytic leukaemia cells [84], mantle cell lymphoma cells [98], follicular lymphomas [99], primary effusion lymphomas [100] and small-cell carcinomas of the lymph nodes [101].

#### 2.3.2. Digestive System

MCPyV sequences have been detected in various tumour samples from the digestive tract. Amongst the different cancers, oesophagus cancer (156 specimens), liver cancer (27 specimens) and salivary gland cancer (185 specimens) seem to have the highest prevalence of the virus, reaching up to 62%, 45.1% and 26.2%, respectively [76,82,102,103]. The detection was performed with the use of qPCR assays. In contrast, qPCR analyses of gallbladder [37], pancreas [37], intestine [106,107], appendix [106] and gastrointestinal cancers [108] indicated no trace of MCPyV genome. Another examined digestive-tract-associated cancer is colon cancer. Colon cancer is the third on the list of cancer-associated fatalities in Western countries, estimated to have caused 9% among all cancer deaths in 2020. Some evidence points to pathogens, such as viruses, as risk factors for its oncogenesis. [159,160,161]. Nevertheless, studies showed a very low prevalence (3.5%) with low and even no positivity toward viral transcripts [82,109,110,111,112,113]. The same pattern was observed in stomach cancer (1.7% prevalence) [82,109,113]. Regarding the upper regions of the digestive system, the oral cavity, some viral genome copies have been detected there. Lastly, some MCPyV-positive tumours were identified in oral cavity cancers; LT protein was not detected in any of them, and the copy number of the virus/cell was low (0.00024–0.026) [82,102,104,105,114,115].

#### 2.3.3. Excretory System

The excretory system is the regulator of the chemical composition of body fluids. It is responsible for the discarding of metabolic wastes of the body, as well as the preserving of the adequate water, salts and nutrients levels within the cells and tissues. Studies have investigated the presence of MCPyV in cancers extracted from the key components of this system, the kidneys [76,109] and the bladder [76,109,113]. In both cases, the prevalence of the virus was low (4% derived from 149 samples of bladder cancer and 3.7% from 81 samples of renal cancer), with a viral load of 0.001 and 0.004 copies/cell [76,82]. Furthermore, IHC assays could not detect any sign of the LT antigen in any of the samples [109,113].

#### 2.3.4. Integumentary System

The integumentary system is the body’s outer layer, meaning it consists of the skin with its glands and nerves, nails, and hair. Since the Merkel cell carcinomas originate from and emerge on the skin, other skin cells in proximity to the tumorigenesis location could potentially be susceptible to the presence of the virus and, consequently, become malignant. Therefore, MCPyV has been searched for and eventually detected in the integumentary system. Initially, MCPyV DNA was detected at a varying level in non-melanoma cancers, acquired from immunocompromised patients (prevalence 31.3%) [82,112]. In this rate are included many non-melanoma skin cancers, such as squamous cell carcinomas (SCC) [112,114,116,117,118,119] and basal cell carcinomas (BCC) [112,120,121,122]. The presence of the virus was also detected in a few cases of keratoacanthoma (viral load 0.0001–0.10 viral genome copies/cell, no LT detected) [82,117,120,123], Kaposi’s sarcoma (0.00001–0.00685 viral genome copies/cell) [82,100,124], porocarcinoma (0.00022–0.212 viral genome copies/cell) [82,114,125] and atypical fibroxanthoma (0.0001–0.031 viral genome copies/cell) [82,120]. On the contrary, melanomas do not seem to typically harbour MCPyV [106,126,127]. There is only one documented case in which acral lentiginous melanomas and in nodular melanomas tested positive [121].

#### 2.3.5. Lymphatic System

In the available data derived from studies on some lymphatic system cancers (hypertrophy adenoid [132]), the prevalence of MCPyV was found to be low. Nevertheless, tonsillar squamous cell carcinoma (Tonsillar SCC) and thymoma qPCR analysis demonstrated a 32% and 15.2% prevalence of the viral DNA, respectively [82,128,129,130,131]. Additionally, three MCPyV DNA-positive thymoma samples also expressed the LT antigen [131]. It is crucial to mention, though, that Toracchio et al. [137] identified transcripts of the other oncogenic antigen of MCPyV, sT antigen, within benign, non-malignant lymph nodes.

#### 2.3.6. Nervous System

Studies focusing on tumours of the nervous system are scarce. Only a limited number of cases in which the MCPyV genome was detected are available and they include brain tumours [101,109,133,134,135,136], schwannomas [102,115,137], a rare type of nerve sheath tumour, mostly benign, meningiomas [102] and glioblastomas [102,138]. Furthermore, Matthay et al. analysed the viral presence in neuroblastomas, the most common malignant extracranial solid tumour in childhood [139]. The results indicated no correlation between the viral DNA in the neuroblastoma samples [113,139,140,141].

#### 2.3.7. Respiratory System

As mentioned already, MCPyV has been detected on respiratory samples and lung fibroblasts are an eligible cell system for the in vitro viral reproduction [34]. Therefore, a lot of hypotheses have arisen regarding the possibility of a linkage between MCPyV and carcinogenesis events in the respiratory system. Additionally, healthy lung tissues have been examined regarding the presence of the virus, and genome transcripts have been detected [37,76,100,142]. Therefore, much research has focused on the detection of genome transcripts in respiratory malignancies. Initially, MCPyV was detected via qPCR and nPCR in lung cancer only in 10 positive cases out of the 388 tested samples, and IHC did not show any detectable level of LT antigen expression in any of the positive samples [37,76,142]. Similarly, the majority of the studies on small-cell lung carcinomas point out a general lack of viral presence, with only a few tumours being MCPyV DNA-positive (5.2% prevalence with 0.000005–0.026 viral copies per cell) [82,107,113,143,144,145,146]. In contrast, scientific evidence has led to a strong association between non-small-cell lung carcinoma pathogenesis and the presence of MCPyV (16.3% prevalence) [82,130,147,148,149]. Even though most of the small-cell lung carcinoma samples showed no LT expression [150], two of the ones that did are of exceptional significance. In particular, those two samples contained a truncated LT antigen and one of those seemed to express both the full-length and truncated LT protein [130]. Further investigations on this specific, peculiar sample, led to the finding of two viral DNAs in the same malignancy sample, both in episomal and integrated form. A similar observation was made in a few Merkel cell carcinomas, where episomal and integrated viral genomes co-existed in the cytoplasm and nucleus simultaneously [130]. Additionally, studies have observed an association between the presence of Merkel cell polyomavirus with a deregulated expression of two genes (BRAF and Bcl-2) in non-small cell lung cancer [151], as well as virus-specific microRNA signature [152].

#### 2.3.8. Reproductive System

The data regarding the involvement of MCPyV in tumorigenesis in the human reproductive system are inadequate. However, there are studies focused on the detection of the virus in malignant samples acquired from reproductive organs that demonstrate some trace of viral presence both in females and males. The samples tested originated from prostate [76,153] and testicular cancer in males [76] (the two most common male-reproductive system cancers) and cervical [155,156], breast [157], ovarian and vulvar cancer [156,157] in women. Initially, MCPyV was detected via qPCR (LT, VP1 sequences) in one case of testicular cancer, a very common, highly treatable, and usually curable cancer in men [76]. Interestingly, the viral load was relatively high, at 0.934 copies/cell [76,82]. The LT protein expression, though, was not assessed [76]. Furthermore, MCPyV-positive prostate cancers, the second-most common type of cancer and the fifth-leading cause of cancer-related death in men [154], have been analysed with a variety of methods for both the viral genome and the LT antigen (qRT-PCR, NGS, IHC). The viral load was found to be lower than in the testicular cancer case, at 0.002 copies/cell, whereas no LT antigen was detected in the tumour [76]. Regarding the female reproduction system, breast cancer is the most prevalent cancer in the female population worldwide [158], with an estimated 1.38 million new cases and over 450,000 deaths annually. The aetiology of the minority of breast cancers is genetic-related, though the rest of the sporadic breast cancers remain a mystery. Studies have proposed a possible aetiological relationship between viral infection breast cancer oncogenesis, which remains to be established. Hence, approximately 474 samples of breast cancer tissue have been used for MCPyV genome detection, via PCR along with whole transcriptome sequencing [82]. Out of all samples, viral genomic transcripts were identified in only a few cases of breast cancer cells [113,157]. Furthermore, other research studies have attempted to determine the prevalence of MCPyV in cervical cancer (the third most prevalent malignancy and the fourth most fatal cancer in women [3,162]) as well as to assess a possible connection between MCPyV and HPV-related cervical cancer. The results demonstrated that the cervical cancer samples contained a relatively low viral copy number per tumour cell, between 0.00003055 and 0.0015 [82,155,156]. Despite the low viral genome quantity, LT antigens (transcripts and proteins) were detected superficially in samples isolated from HIV-positive women [156]. Finally, MCPyV viral transcripts were not detected in either ovarian or vulvar cancer samples [113,140].

#### 2.3.9. Soft Tissue

Lastly, soft tissue is all the tissue in the body that is not ossified or calcified, and, therefore, remains unhardened. The role of soft tissue is the connection, surrounding and support of other tissues, internal organs, and bones. This tissue category comprises muscles (including the heart), fat, nerves, as well as lymph and blood vessels, ligaments, tendons, and tissues that surround the bones and joints [163,164]. Soft tissue cancers can arise at any age, with an increased prevalence amongst children and young adults, along with adults over the age of 55. Thus far, the soft tissue-related tumours examined (all by Sastre-Garau et al. [140]) are a limited number of rhabdomyosarcoma tumours (25 samples, a rare type of cancer that forms in skeletal muscle tissue), desmoplastic tumours (24 samples, these grow in the abdomen and pelvic area of the body) and fibromatosis tumours (4 samples, a connective tissue malignancy developing in musculoaponeurotic tissues). The detection of the viral sequences was performed by PCR (primers for the conserved LT sequences). The results demonstrated that none of the soft tissue malignant specimens harboured any MCPyV DNA.

## 3. Conclusions

In summary, a causality connection between infectious agents and carcinogenesis is well established. Merkel cell polyomavirus (MCPyV), a small DNA virous, is, thus far, the only polyomavirus associated with oncogenesis in humans. Normally, it is a widespread member of the skin microbiome. However, in some rare cases, especially in immunocompromised individuals, it evidently triggers the pathogenesis of Merkel cell carcinomas (MCC) on the skin. MCC is a rare, extremely aggressive neuroendocrine neoplasia of the skin, of uncertain cell origin, that emerges mainly on UV-exposed areas. It is characterized by low survival rates, rapid metastatic ability, and ever-increasing incidence. Additionally, the available therapeutic approaches are considered inadequate. The data regarding the biology of the virus are insufficient and the precise oncogenic mechanism remains to be elucidated. However, the MCC onset is confirmed to be dependent on two events: viral DNA integration into the host genome and viral oncoprotein expression, such as a truncated version of the LT antigen. Both events, along with other UV-related mutations, lead to interference with the normal cellular life circle and cell proliferation and, consequently, lead to cell transformation.

MCPyV is not detected solely in skin samples, but also in other tissues of the human body. This finding, along with the high seroprevalence of MCPyV in the human population, the in vitro oncogenic potentials of LT and sT, as well as its strong tumorigenic properties on MCC, have triggered attempts to detect MCPyV (DNA and proteins) in non-MCC tumour samples. In total, many of the samples tested displayed the presence of the virus in various non-MCC cancers. Nevertheless, the results do not clearly demonstrate a direct connection between cellular transformation and the presence of the virus, due to some pitfalls of the studies [82]. Firstly, for the virus-positive cases, the numbers of the same tumour types tested are insufficient for a conclusion to be made, and many of the studies were case reports. For statistical significance, large cohort studies are needed. Furthermore, the tested samples were extracted from patients of different health backgrounds, for instance both immunocompetent and immunocompromised patients, and the abstracted results were considered, in many cases, as one category, even though the condition of the immune system of an individual plays a crucial role in the onset and progress of malignant tumours. Additionally, even with the presence of the viral DNA, the prevalence, viral load, and protein expression were significantly lower compared to MCCs and the viral load (genome copies/cell) was scarcely determined. Moreover, the two hallmarks of MCPyV MCC, namely the state of the viral genome and whether a truncated large T antigen is expressed, have not systematically been investigated in non-MCC tumours. Expression of truncated LT and/or integration of the viral genome was only reported in a few cases. Lastly, in several reported cases, the prevalence in healthy tissue was comparable to malignant tissue.

Regarding the virus-negative tumours, the lack of the viral genome within a tumour cannot eliminate the viral–oncogenesis possibility. Initially, the absence of the virus from the tested samples could be attributed to dysfunction of molecular techniques used for the detection. In particular, PCR-based methods were the choice of the vast majority of the research. The results can be affected by the primers that were used, the quality of the samples, as well as possible mishandling and contamination (considering MCPyV is chronically shed from the skin [32]). Other more sensitive techniques such as NGS could provide a highly sensitive approach to detect the viral genome in non-MCC tumours. Furthermore, the total absence of the viral genome does not necessarily exclude a possible scenario where the viral presence is not needed for the tumour progression, as it has been described before. In this scenario, MCPyV would be necessary for tumour initiation, but at later stages the virus is dispensable. In some MCC cases, this “hit-and-run mechanism” for MCPyV, has been suggested based on the observation that a knockdown LT antigen in the MCPyV-positive MKL-1 cell induced growth repression, whereas no impaired growth was reported in LoKe cells that were LT-deprived [165]. Lastly, the viral proteins, without the presence of its genetic material, could have enhancive effects on other co-infecting oncovirus, since the co-presence of MCPyV and human papillomavirus and Epstein–Barr virus in tumours has been reported [129,153,166,167].

In conclusion, MCPyV is the major oncogenic factor for MCCs, but no establishing results can be derived about the virus’ prevalence in non-MCC skin abnormalities and its potential role in the pathogenesis of other malignant diseases. Most certainly, it is imperative that the biochemical, molecular, and immunological characteristics of the virus are further investigated. The more understanding we have on the virus’ biology and infection, the easier it will be to investigate its correlation with the tumorigenesis of other malignant cells, and, also, many more possible preventative and therapeutic targets will be found, taking us one step closer to the elimination of Merkel cell carcinomas.

## Figures and Tables

**Figure 1 cancers-14-05548-f001:**
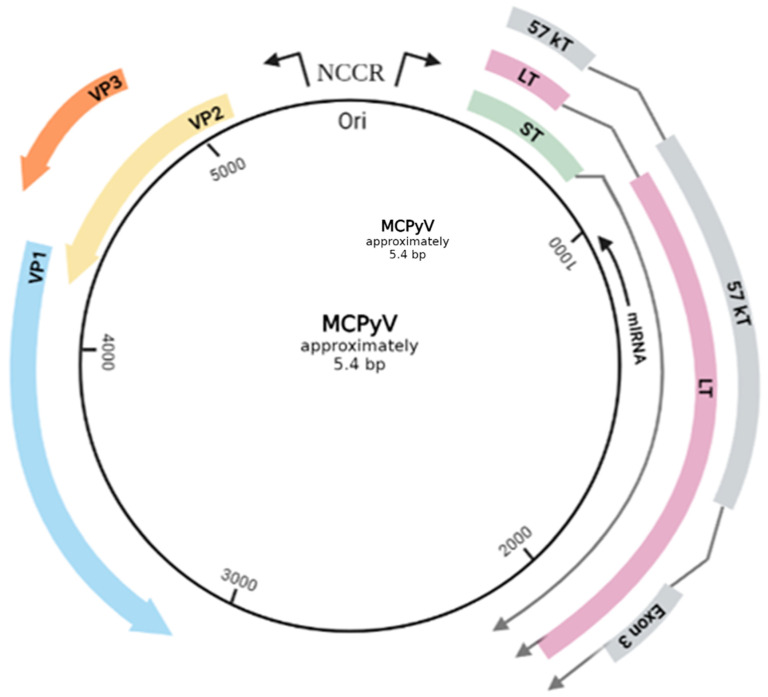
The genomic map of MCPyV circular, double stranded DNA. The early genes encode oncogenic proteins (LT, sT, 57 kT) and the late genes structural proteins (VP1. VP2, VP3). The NCCR (non-coding control region) includes Ori and regulates the viral gene expression. A gene, located on the late strand of the T antigen genes, with an opposite transcription orientation (reversed complementarity) to the T antigens produces a non-coding miRNA. The total length of the viral genome is approximately 5.4 bp. Created with BioRender.com (visited on 1 July 2022).

**Figure 2 cancers-14-05548-f002:**
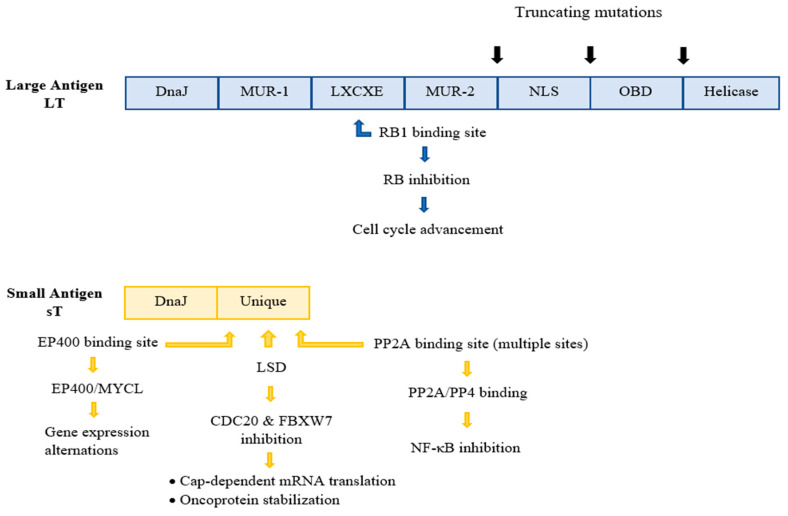
The structural domains of the early gene region antigens, LT and sT. The LT and ST are products of alternative splicing of the region. In MCC, a truncated version of LT is expressed, characterized by mutations that eliminate domains of the C-terminus of LT, naming the nuclear localization signal (NLS), origin binding domain (OBD), and helicase domains. The crucial for the oncogenesis LXCXE RB-binding motif is preserved. ST contains several unique motifs with multiple sites that bind and interact with a variety of host cellular proteins. The DnaJ domains, conserved in both antigens, are related to viral replication. NLS, Nuclear localization signal; MUR, MCPyV unique region; OBD, Origin binding domain, PP2A protein phosphatase 2A; EP400, E1A-binding protein p400; MYCL, L-myc-1 proto-oncogene protein; LSD, NF-κΒ, Nuclear factor kappa B; LT-stabilizing domain; FBXW7m F-box/WD repeat-containing protein 7; CDC20, cell division cycle protein 20 homolog; PP4, Protein phosphatase 4.

**Table 1 cancers-14-05548-t001:** Detection results and method of the MCPyV DNA in tumours derived from different tissues. For the positive cases of the same type of tumours tested, the prevalence of the viral DNA varies greatly.

Histological Tumour Type	System	MCPyV DNA	Method	Complementary Findings	References
AML	Circulatory System	+	qPCR, NGS	Viral DNA found in only 1 sample	[82,83,84]
CTCL	Circulatory System	+	PCR/Southern blot, qPCR	-	[82,85,86,87,88]
CBCL	Circulatory System	+	PCR/Southern blot, qPCR	All samples tested by IHC were negative for LT	[82,88,89,90]
CLL/SLL	Circulatory System	+	PCR, qPCR, FISH, IHC	CLL patients have a higher risk of developing VP-MCCs. six samples contained a truncated LT antigen two of the above also harboured full-length LT mRNA	[82,91,92,93,94,95,96,97]
Chronic myelomonocytic leukemia	Circulatory System	−	qPCR	-	[84]
		−	FISH	-	[98]
Mantle cell lymphoma	Circulatory System				
Follicular lymphoma	Circulatory System	−	PCR, qPCR	-	[99]
Primary effusion lymphoma	Circulatory System	−	qPCR	-	[100]
Small-cell carcinoma of the lymph nodes	Circulatory System	−	IHC	-	[101]
Esophagus cancer	Digestive System	+	qPCR	-	[76,82,102]
Liver cancer	Digestive System	+	qPCR, IHC	-	[82]
Salivary gland cancer	Digestive System	+	qPCR, IHC	In one positive parotid sample truncated LT was detected	[82,102,103,104,105]
Gallbladder cancer	Digestive System	−	qPCR	-	[37]
Pancreas cancer	Digestive System	−	qPCR	-	[37]
Intestine cancer	Digestive System	−	nPCR	-	[106,107]
Appendix cancer	Digestive System	−	IHC	-	[106]
Gastrointestinal cancer	Digestive System	−	IHC	-	[108]
Colon cancer	Digestive System	+	PCR, nPCR, qPCR, FISH, IHC	-	[82,109,110,111,112,113]
Stomach cancer	Digestive System	+	PCR, IHC	-	[82,109,113]
Oral cavity cancer	Digestive System	+	PCR, qPCR, IHC	-	[82,102,104,105,114,115]
Bladder cancer	Excretory System	+	qPCR, IHC	-	[76,82,109,113]
Renal cancer	Excretory System	+	qPCR, IHC	-	[76,82,109,113]
SCC	Integumentary System	+	PCR, nPCR, qPCR, ddPCR, NGS, IHC	All samples tested by IHC were negative for LT Truncated LT was found in some samples after sequencing	[112,114,116,117,118,119]
BCC	Integumentary System	+	PCR, nPCR, IHC, nPCR,	All samples tested by IHC were negative for LT	[112,120,121,122]
Keratoacanthoma cancer	Integumentary System	+	PCR, nPCR, qPCR	All samples tested by IHC were negative for LT	[82,117,120,123]
Kaposi’s Sarcoma	Integumentary System	+	PCR, nPCR, qPCR	-	[82,100,124]
Porocarcinoma	Integumentary System	+	PCR, nPCR, qPCR, ddPCR	-	[82,114,125]
Atypical fibroxanthoma	Integumentary System	+	PCR, qPCR, IHC, PCR/Southern blot	All samples tested by IHC were negative for LT	[82,120]
Melanoma	Integumentary System	−	PCR, nPCR	only 1 documented case, in which acral lentiginous melanomas and nodular melanomas were tested positive	[106,121,126,127]
Tonsillar SCC	Lymphatic System	+	qPCR	-	[82,128,129,130,131]
Hypertrophy adenoid	Lymphatic System	+	qPCR	-	[132]
Brain tumours	Nervous System	+	qPCR, nPCR, IHC	-	[101,109,133,134,135,136]
Schwannomas	Nervous System	+	qPCR	One examined sample contained LT transcripts	[102,115,137]
Meningiomas	Nervous System	+	qPCR	-	[102]
Glioblastomas	Nervous System	+	qPCR	-	[102,138]
Neuroblastomas	Nervous System	-	qPCR	-	[113,139,140,141]
Lung cancer	Respiratory System	+	PCR, qPCR	-	[37,76,107,117,142,143]
SCLC	Respiratory System	+	PCR, qPCR, nPCR, IHC	-	[82,107,113,143,144,145,146]
NSCLC	Respiratory System	+	PCR, qPCR, RT-PCR	Integrated viral DNA in one adenosarcoma + expression of truncated LT. Integrated + episomal viral DNA in one SCC sample + expression of full-length and truncated LT association between the presence of Merkel cell polyomavirus with a disrupted expression of two genes (BRAF and Bcl-2) as well as virus-specific microRNA signature	[82,130,147,148,149,150,151,152]
Prostate cancer	Reproductive System	+	qPCR, qRT-PCR, NGC, IHC	-	[76,153]
Testicular cancer	Reproductive System	+	qPCR, nPCR	-	[76,154]
Cervical cancer	Reproductive System	+	PCR, qPCR, RT-PCR, IHC	140 of the examined patients were also HIV positive	[82,155,156]
Breast cancer	Reproductive System	+	PCR, qPCR, qRT-PCR	-	[82,157,158]
Rhabdomyosarcoma tumours	Soft Tissue	−	PCR	-	[140]
Desmoplastic tumours	Soft Tissue	−	PCR	-	[140]
Fibromatosis tumours	Soft Tissue	−	PCR	-	[140]

Acute Myeloid Leukemia (AML); Cutaneous T-cell leukemia (CTCL); Cutaneous B-cell lymphomas (CBCL); Chronic Lymphocytic Leukemia/Small Lymphocytic Lymphoma (CLL/SLL); Squamous Cell Carcinomas (SCC); Basal Cell Carcinomas (BCC); Tonsillar Squamous cell carcinoma (Tonsillar SCC); small-cell lung carcinomas (SCLC), non-small-cell lung carcinomas (NSCLC).

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
