# Peer review of "Merkel Cell Polyomavirus (MCPyV) and Cancers: Emergency Bell or False Alarm?"

_cancers, 2022, doi:10.3390/cancers14225548_

Round 1
Reviewer 1 Report
Generally I think it is a nice idea to summarize all the findings on MCPyV detection in different cancers. In this respect the review provides a comprehensive overview on, however, frequently only very weak data presenting studies . The latter is of course not attributable to the authors of the review. However, when reading the abstract and those parts of the text describing the current knowledge on MCPyV and the functions of its oncoproteins, there are several inconsistencies. I suggest once again studying the literature and then improving the text accordingly
Some specific issues:
Abstract
“Though, it is established that oncogenic transformation is the outcome of two events: 1) the integration of the viral genome 2) the expression of a truncated version of the Large-T oncoprotein that interferes with regulatory pathways of the cell cycle. “
This statement is wrong; MCPyV small T antigen plays a major role and many results suggest that LT is less important or even dispensable for transformation
“along with the wide spread of the virus across the human body”
I don’t understand what the authors want to suggest with this phrase
Page 3 line 98: it sounds as if MCPyV LT would inactivate p53 which is not the case
Figure 1: Looks like the miRNA is encoded in the early region which is not the case
Figure 2: As in the text, also in this figure MCPyV-sT is not adequately described . Indeed, the PP2A interaction has been suggested not to be crucial for its function while several other novel functions, not described for other polyomavirus sT proteins, have been revealed. Also the depiction of LT is to schematic not representing
Table 1 is missing and could, therefore, not be evaluated. This table, however, can be the most important part of the review.
Line 332: an “of” is missing
Line 397: “disrupted expression” is an ambiguous description
Line 399 – 401: in my view the given conclusion has no basis in the observations made
I think the summary is too long and I would suggest to provide a more concise version.
Author Response
Reviewer No 1
Generally, I think it is a nice idea to summarize all the findings on MCPyV detection in different cancers. In this respect the review provides a comprehensive overview on, however, frequently only very weak data presenting studies. The latter is of course not attributable to the authors of the review. However, when reading the abstract and those parts of the text describing the current knowledge on MCPyV and the functions of its oncoproteins, there are several inconsistencies. I suggest once again studying the literature and then improving the text accordingly
Some specific issues:
Abstract
“Though, it is established that oncogenic transformation is the outcome of two events: 1) the integration of the viral genome 2) the expression of a truncated version of the Large-T oncoprotein that interferes with regulatory pathways of the cell cycle”.
This statement is wrong; MCPyV small T antigen plays a major role and many results suggest that LT is less important or even dispensable for transformation
“along with the wide spread of the virus across the human body”
I don’t understand what the authors want to suggest with this phrase
Au: We would agree with the reviewer’s comment regarding the misleading of the statement which is due to the misuse of words from our side. Indeed, the main, well-established event that proceeds the oncogenic transformation on viral MCCs is the integration of the viral genome into the host DNA. However, the second statement is not a hallmark, rather than a very common characteristic of MCPyV positive MCCs: the expression of a truncated version of one of the oncoprotein LT antigen. The abstract has been modified, accordingly. Regarding the line “along with the wide spread of the virus across the human body”, as mentioned in the text later on (lines 141-147), the virus is not solely detectable the skin, but also in other tissue samples (respiratory samples, urine, and blood) as well as in samples extracted from numerous non-malignant tissues. This line aims to highlight this wide spread of the virus across the body and not just the skin. The current line has been rephrased to hopefully, better depict the meaning.
Page 3 line 98: it sounds as if MCPyV LT would inactivate p53 which is not the case
Au: According to the reference used (Wendzicki JA et al. 2015, doi:10.1016/j.coviro.2015.01.009) “In the context of oncogenesis, some of these elements (LT antigen motifs) also have the effect of disabling tumor suppressor pathways, for example by targeting Rb and p53.” Therefore, we maintained the reference’s message and no alternations have been made on this sentence (line 111).
Figure 1: Looks like the miRNA is encoded in the early region which is not the case
Au: The text, as well as the figure memorandum have been updated in order to clarify that the miRNA is expressed from the late strand which is reverse complementary to the early coding genes (lines 99-101, 122-125).
Figure 2: As in the text, also in this figure MCPyV-sT is not adequately described. Indeed, the PP2A interaction has been suggested not to be crucial for its function while several other novel functions, not described for other polyomavirus sT proteins, have been revealed. Also the depiction of LT is to schematic not representing
Au: The main aim of our review article is to summarize the studies about the presence of the virus in a variety of cancer types. We thought it was necessary, in order to be well organized, complete and easily comprehended by everyone, to firstly introduce the virus, hence we included its characteristics, in no high detail. The reason why sT was not extensively described throughout the text is because, the vast majority (if not all) of the studies used for the article identify the viral presence either by the detection of the viral genome or/and the detection of LT antigen in the tested samples, and not sT. However, we took into consideration the reviewer’s comment and we did include more functions of sT in the text, as suggested (lines 244-258). Regarding the Figure, a few alternations have been made to fit your request (lines 269-281), however, if you still find it highly inadequate, we opt for completely removing it from the article.
Table 1 is missing and could, therefore, not be evaluated. This table, however, can be the most important part of the review.
Au: We apologize for the omission and the inconvenience. The Table has been added properly after line 330, page 8.
Line 332: an “of” is missing
Au: It has now been inserted (line 393).
Line 397: “disrupted expression” is an ambiguous description
Au: The word was substituted by “disregulated expression” in the revised manuscript, which is more suitable, perfectly describing the reference (line 458).
Line 399 – 401: in my view the given conclusion has no basis in the observations made
Au: After revision and careful consideration, the specific point has been deleted from the text (line 460-462).
I think the summary is too long and I would suggest to provide a more concise version.
Au: We took into consideration the reviewer’s comment and the summary has been modified to be more concise, as suggested.
Reviewer 2 Report
This review entitled “Merkel Cell Polyomavirus (MCPyV) and cancers: emergency bell or false alarm?” submitted by Maria Georgia Dimitraki et al. was generally well organized. The authors started from the oncogenic viruses and focused on MCPyV and MCC, and further reviewed the literature and summarized the prevalence of MCPyV in different human systems and raised a question linking the presence of this virus and the non-MCC tumors. The conclusion is that no established results can be derived about this virus prevalence in non-MCC skin abnormalities, and no potential roles of this virus in the pathogenesis of other malignances so far, since the cases number limitation and some other reasons.
There is three small point should be raised: 1) when the author described the Merkel Cell Polyomavirus, you mentioned the early transcript and the multiple gene products (line 83-87) and you also included the virus encoded miRNA. Actually, this miRNA gene is located in the negative strand of T antigens, its transcription orientation is totally opposite to the T antigens (as indicated in Figure 1), it cannot be transcribed with T antigens together. You should clarify and correct it; 2) In figure 1, the authors provide a scheme showing the genomic map of MCPyV, and labelled the full length as 5386 bp. Actually, NCBI presented different reference sequences for MCPyV, the length of them is a little bit different, I would suggest you label the reference sequence number or give an approximate length; 3) the Table 1 (after the line 278) is missing.
Author Response
This review entitled “Merkel Cell Polyomavirus (MCPyV) and cancers: emergency bell or false alarm?” submitted by Maria Georgia Dimitraki et al. was generally well organized. The authors started from the oncogenic viruses and focused on MCPyV and MCC, and further reviewed the literature and summarized the prevalence of MCPyV in different human systems and raised a question linking the presence of this virus and the non-MCC tumors. The conclusion is that no established results can be derived about this virus prevalence in non-MCC skin abnormalities, and no potential roles of this virus in the pathogenesis of other malignances so far, since the cases number limitation and some other reasons.
There is three small point should be raised:
1) when the author described the Merkel Cell Polyomavirus, you mentioned the early transcript and the multiple gene products (line 83-87) and you also included the virus encoded miRNA. Actually, this miRNA gene is located in the negative strand of T antigens, its transcription orientation is totally opposite to the T antigens (as indicated in Figure 1), it cannot be transcribed with T antigens together. You should clarify and correct it;
Au: The information was apparently not clear to the reviewer (although Figure 1. depicts it correctly). Considering the reviewer’s comment, we have rephrased the text in the revised manscript. (line 97-101).
2) In figure 1, the authors provide a scheme showing the genomic map of MCPyV, and labelled the full length as 5386 bp. Actually, NCBI presented different reference sequences for MCPyV, the length of them is a little bit different, I would suggest you label the reference sequence number or give an approximate length;
Au: After the reviewer’s comment we revised Figure 1., as well as the text, and substituted the genome length with an approximate number, in order to include all the small length differences presented in different references.
3) the Table 1 (after the line 278) is missing.
Au: We apologize for the omission and the inconvenience. The table has been added right after line 330.
Round 2
Reviewer 1 Report
The authors have been sufficiently responsive to my comments.